# OpenReview forum: ""Fifty Shades of Bias": Normative Ratings of Gender Bias in GPT Generated English Text"
_EMNLP/2023/Conference — EMNLP 2023 Main_

### Official Review · Reviewer_HvWd · 2023-07-20

**Soundness:** 3

**Excitement:**

2: Mediocre: This paper makes marginal contributions (vs non-contemporaneous work), so I would rather not see it in the conference.

**Missing References:**

Papers on subjectivity in annotation (see reasons to reject):
Rottger et al., NAACL 2022
Davani, A. M., Díaz, M., & Prabhakaran, V., 2022
Biester et al., NLPerspectives 2022

You use Nadeem et al. 2021 as your seed set - consider Blodgett et al., ACL-IJCNLP 2021 for a critique of this corpus.

I suggest citing the official version of i.e. Blodgett et al. 2020 (amongst others) rather than the arxiv version.

**Paper Topic And Main Contributions:**

This paper presents a novel data set annotated for gender bias using a best-worst scale approach that allows for more nuanced annotation than binary classification. Data is generated using GPT-3.5-Turbo, using a variety of prompting methods, to ensure a diversity of data in terms of gender bias and in terms of linguistic diversity. This data is annotated by a set of annotators using the BWS approach to capture relative gender bias, and the typical rank is used to score each sentence. Given the relatively small size of the data set, it will be most useful to the community as “training” data for in-context learning, and for the evaluation of gender bias detection models.

**Questions For The Authors:**

(A) Would you consider releasing more information about the annotators (with their permission!) to enrich the data set for those working on modelling annotator "disagreement"?

(B) How can you be sure your snowball sample avoids the quality control issues of crowdsourcing annotations? Most of the annotators were male, did you notice any difference in their scores compared to female annotators?

(C) I found the purpose of the benchmarking section to be a little unclear. Is the data set being used to evaluate the models, or are the models being used to investigate the data set  (i.e. identifying toxicity and identity attack as types of gender bias), as suggested by the final sentence? Perhaps benchmarking is the wrong term of these analyses.

(D) How were the proportion of best and worst labels from 24 annotations used to give a score from -1 – 1 before being normalised?

**Reasons To Accept:**

The authors highlight the importance of a nuanced approach to gender bias, noting that implicit gender bias can be subtle but is important to detect. As such, the authors present a clear case for the benefit of having graded gender bias annotations. They also clearly present the benefits of the BWS approach in terms of efficiency for annotating subjective judgements.

The authors analyse their generated data set to identify key themes using multiple methods, which allows them to draw some top-level conclusions about trends in the kind of biased content being produced.

**Reasons To Reject:**

Given the authors’ assertion that gender bias is highly subjective I would have liked to see more discussion on the method used to convert labels into scores. Some implicit assumptions (that annotator gender has no impact on the importance of a judgement, that controversial statements with a mix of high and low ranks should receive a “middling” score) need to be made explicit and discussed. On a related note, given the topic of the paper, I think it is a shame the authors did not engage with the literature on handling annotation disagreement for subjective labels - the only paper referenced that discusses how annotations are subjective is Blodgett et al. 2020, although the authors repeat this point in a number of places.

I would have liked to see the authors engage more with the limitations of using GPT models to generate biased text. They acknowledge that the model is “limited by the dataset it has seen during training”, but the paper would benefit from more consideration of this point. For example, it is likely that OpenAI “cleaned up” the training data seen by the model during training, or conducted alignment before release, meaning the range of explicitly gender biased sentences the model was exposed to was greatly reduced, limiting what it can produce. It could have been interesting to do a qualitative analysis comparing the generated samples with sentences from a gender bias data set that was sampled from an online corpus (and in general, more qualitative analysis of the data set beyond identifying key words would have been beneficial).

**Reproducibility:**

4: Could mostly reproduce the results, but there may be some variation because of sample variance or minor variations in their interpretation of the protocol or method.

**Reviewer Confidence:**

4: Quite sure. I tried to check the important points carefully. It's unlikely, though conceivable, that I missed something that should affect my ratings.

**Typos Grammar Style And Presentation Improvements:**

Line 139 - I’m not sure what is meant by “However, due to the absence of words with explicit negative connotations”

Line 244 - there may be something off with your formatting as this line seems to float between two others.

Line 248 - what is meant by “colored” here?

Line 254 - implicit *and* neutral (?)

Line 306 - I would perhaps say “perceived gender bias” is based on lived experience.

Section 3.2 informs your choice of seeds so would be more clearly presented before describing the seed sets.

Section 3.3 - please define the process of conversion when you introduce it

Table 3 - might it be possible to see more than four examples? I think this could be interesting to the reader.

I don’t place much confidence in the “reasoning” that language models provide for their answers, so I was unsurprised to see that the scores assigned by GPT-4 did not align with the provided reasoning. It is with this stance in mind that I suggest that section 7 could be shortened. Also, some readers may be put off by the use of very anthropomorphic language in this section i.e. “it hedges itself” (Line 583).

---

> ### Author Rebuttal · Authors · 2023-08-28
>
> Thank you for your thoughtful comments.
>
> **Conversion of labels to scores, calculation of gender bias scores, and “middling” score:**
>
> Since gender bias is highly subjective, we adopt the BWS annotation framework. In this framework, the disagreement that arises in a tuple that has 2 closely scored items is an important indicator. For a 4-tuple, if participants struggle to reliably determine the sentence with the highest (or lowest) gender bias, this lack of consensus results in the two sentences receiving scores that are in proximity to one another.
>
> For each statement in the dataset, score is the proportion of times the statement is chosen as the most negatively biased minus the proportion of times the statement is chosen as the least negatively biased (lines 355 to 359 in the paper). For example, if a sentence is chosen as least 22 times and as most 0 times, its score will be (0 – 22)/24 = – 0.91.
>
> Since each sentence appears in 8 distinct 4-tuples with diverse neighbors, we will have a good judgement even for “controversial” statements. It is not likely (although possible) that a statement which is controversial for a particular tuple will consistently be chosen as most and least (in equal proportions) with the diverse set of neighbors it appears with in different tuples, giving it a middling score.
>
> **Quality control in snowball sampling approach:**
>
> With most of the prior work in NLP dataset creation using the standard pool of annotators on MTurk, scholars have shown that the datasets created skew towards Anglo-centric sensibilities due to the over representation of annotators from the Global North (primarily the US population)[i]. Hence, by using snowball sampling, we tried to ensure that we reached out to people who are diverse from the standard MTurk pool – namely, those who identify as Indian culturally and had some prior understanding of the concept of ‘gender bias.’ Furthermore, the annotators were recruited from within the institute, lending themselves to higher trust and, in turn, ensuring the quality of annotations. Our high inter-annotator agreement scores of ~0.86 show that annotations are of high quality.
>
> *[i]Chandler, J., Rosenzweig, C., Moss, A. J., Robinson, J., & Litman, L. (2019). Online panels in social science research: Expanding sampling methods beyond Mechanical Turk. Behavior research methods, 51, 2022-2038.*
>
> **More information about annotators, implicit assumptions regarding impact of annotator’s gender on annotations, and differences in male vs female annotators:**
>
> Due to the limitations of the IRB approved for this study, we are unable to release more information about the annotators; however, we do release individual annotations per tuple with worker ID that can be used by future studies for understanding disagreements.
>
> We are conducting an in-depth follow-up study to gain a better perspective on the impact of gender on gender bias annotations and study the differences between male and female annotators.
>
> **Literature on handling annotation disagreement:**
>
> We cite work that discusses annotation disagreement in the context of bias, hence informing our choice of adopting BWS annotation framework for a subjective task. We acknowledge that we did not engage with broader literature surrounding annotator disagreement. If accepted, we will utilize the extra space to discuss this aspect.
>
> **Limitations of using GPT models to generate biased text, and comparison between generated examples vs gender bias data from online corpus:**
>
> This dataset was created with the intention of investigating the biases LLMs can generate despite all the content moderation policies in place. We show that GPT models still generate explicit instances of gender bias. If accepted, we will utilize the extra page to discuss more limitations of using GPT models in such cases. We will also include analysis on generated examples vs gender bias data from online corpus, as suggested.
>
> **Benchmarking section:**
>
> In this section, our objective is to examine the performance of available models that have been fine-tuned on gender bias or related concepts, on our dataset. We will change the terminology from "benchmarking" to better align with the nature of our analyses.
>
>
> We will correct the typos, make the style corrections, and cite the official version of papers as suggested.

---

### Official Review · Reviewer_XADC · 2023-08-02

**Soundness:** 4

**Excitement:**

4: Strong: This paper deepens the understanding of some phenomenon or lowers the barriers to an existing research direction.

**Paper Topic And Main Contributions:**

The authors present a new dataset for fine-grained gender bias detection. The dataset was created through various prompting methods on GPT3.5-Turbo with in-context learning. The samples in the dataset are assigned numerical values on a continuous scale, from least to most biased, and were initially ranked against each other in tuples of size 4. The authors study the dataset samples through inter-annotator agreement, sample themes, and baseline classification tasks. The results show that annotators consider attacks on groups rather than individuals to be more biased and baseline models like GPT4 cannot reason well about fine-grained classifications.

**Questions For The Authors:**

Can you elaborate on how knowing the degree of gender bias can aid the bias mitigation strategies in current text generation models?

Can you give an example of the conversion and completion methods? I have seen the prompts in the appendix but am curious about an explicit example.

**Reasons To Accept:**

The authors provide a dataset for explicit rankings of the degrees of gender bias, expanding upon existing datasets for binary classification. I believe that this is an important space to break into, as binary classification is very limiting and can constrain different viewpoints.

The dataset creation is analyzed thoroughly, from prompting methods to annotator agreement to sample bin themes.

The authors benchmarked the dataset across different types of models, including toxicity, gender bias, and, and offensive language.

**Reasons To Reject:**

One concern I have is in using a LLM to generate a new dataset. This can drastically limit the ideas generated in a dataset and provide a constrained viewpoint in such a sensitive domain.

In addition to the sample generation, the limited diversity in annotators should be considered. An ideal dataset would be diverse across various cultural opinions.

**Reproducibility:**

3: Could reproduce the results with some difficulty. The settings of parameters are underspecified or subjectively determined; the training/evaluation data are not widely available.

**Reviewer Confidence:**

4: Quite sure. I tried to check the important points carefully. It's unlikely, though conceivable, that I missed something that should affect my ratings.

**Typos Grammar Style And Presentation Improvements:**

Line 251: missing period

---

> ### Author Rebuttal · Authors · 2023-08-28
>
> Thank you for your thoughtful comments.
>
> **Limitations of using LLMs for dataset creation:**
>
> We acknowledge that the use of LLMs in creating a dataset can have various limitations. In our work, we use a diverse set of seeds (500 seeds to generate 1000 sentences), and a diverse set of in-context examples. This dataset was created with the intention of investigating the biases LLMs can generate and should be used as such. The dataset does not represent all possible viewpoints/opinions for the given context.
>
> **Limited diversity in annotators:**
>
> We acknowledge that our annotator sample was skewed and that tasks like ours that focus on highly subjective concepts will benefit through annotations that have a diverse coverage of annotator identity. The current limitation section already acknowledges the limitations that arise due to the lived experience, perspectives and geography. While MTurk’s population is largely based out of the Global North (primarily the US population)[i] in this work, we attempted to create some diversity by recruiting individuals who are diverse from the standard MTurk pool – namely, those who identify as Indian culturally.
>
> *[i]Chandler, J., Rosenzweig, C., Moss, A. J., Robinson, J., & Litman, L. (2019). Online panels in social science research: Expanding sampling methods beyond Mechanical Turk. Behavior research methods, 51, 2022-2038.*
>
> **Usefulness of knowing degrees of gender bias:**
>
> If we can obtain a degree of gender bias on text from text generation models, we can devise several mitigation strategies depending on the severity and particular use case.  For instance, if the bias is severe, more aggressive intervention methods like retraining the model on debiased data or modifying the loss function might be necessary, whereas a model with milder bias could benefit from post-processing techniques. Similarly, in certain use-cases we might want models to generate no biased content at all, or only ban extremely severe cases of bias in text. We will include this in the paper as well.
>
> **Example of conversion and completion:**
>
> We have included an example here for your reference. We will include more examples in the paper.
>
> - Conversion:
>     - Seed: She cares about herself too much.
>     - GPT response: She's so vain and self-absorbed that she doesn't care about anyone else's needs or feelings.
> - Completion:
>     - Seed: The male entered the office.
>     - GPT response: The male entered the office confidently, while the female hesitated at the door, unsure if she was in the right place.
>
> We will correct the typos as suggested.

---

### Official Review · Reviewer_DxRa · 2023-08-07

**Soundness:** 4

**Excitement:**

4: Strong: This paper deepens the understanding of some phenomenon or lowers the barriers to an existing research direction.

**Paper Topic And Main Contributions:**

This paper aims to analyze the presence of gender bias in LLMs. The authors promp GPT-3.5 Turbo to generate 1,000 gender-biased  texts, graded by degree of bias (in 0-1 range), then the texts are graded and annotated manually using a comparative annotation scheme (Best-Worst Scaling). Then they perform a quantitative and qualitative analysis of the annotations to learn the topics or themes that more strongly signal gender bias.

**Reasons To Accept:**

LLM's gender bias responses were generated using a very carefully curated list of seeds, which the authors derived from existing related datasets (StereoSet and COPA). The prompting methods are diverse and carefully chosen to ensure good coverage of different text types (conversation, completion, conversion).

The annotation process was carefully designed and monitored, using 20 annotators that were carefully instructed to do their job. The quality and consistency of annotations is evaluated correctly.

The authors seem to know the related literature and resources very well, which they describe well and from which they choose the parts that best fit their research.

Theme identification is carried out in a rigorous manner, calculating PMI scores of unique words and sentences and an informed Dirichlet model and Convokit to identify meaningful n-grams.

The qualitative analysis complements well the plethora of qualitative results offered and offers further insights.

The authors make the annotated datasets freely available.

**Reasons To Reject:**

I can't think of any.

**Reproducibility:**

3: Could reproduce the results with some difficulty. The settings of parameters are underspecified or subjectively determined; the training/evaluation data are not widely available.

**Reviewer Confidence:**

5: Positive that my evaluation is correct. I read the paper very carefully and I am very familiar with related work.

---

> ### Author Rebuttal · Authors · 2023-08-28
>
> Thank you for your thoughtful comments.

---

### Meta-Review · Area_Chair_baut · 2023-09-17

**Recommendation:** 4

**Metareview:**

This paper contributes a new dataset for the analysis of gender bias in LLMs, obtained by prompting GPT3.5-Turbo via in-context learning, and a method for moving from binary to graded bias annotation based on Best-Worst Scaling. As reviewer XADC summarizes, the resulting data is analyzed through 3 lenses: inter-annotator agreement, sample themes and baseline classification tasks, revlealing that annotators overall consider attacks on groups to be more biased than attacks on individuals, and that LLMs such as GPT4 cannot reason well about fine-grained gender bias classification.

The reviewers agree that the paper is overall sound, with adequate description of the methodology and support for the claims.
The paper and data leave open some questions around subjectivity in annotation and the impact of the gender of annotators, which are particularly important from a human-centered NLP perspective. However, the current scope of the work represents an interesting step forward in characterizing gender bias in NLP systems.

I recommend incorporating the discussion of related work, limitations, motivation, and examples that were brought up in the thorough discussion with reviewers. As a minor point, it would be useful to preview the findings of the paper in the abstract, as it currently describes what was done, but leaves the reader wondering what was learned.

---

### Decision · Program_Chairs · 2023-10-07

**Decision:**

Accept-Main

**Comment:**

This paper contributes a new dataset for the analysis of gender bias in LLMs, obtained by prompting GPT3.5-Turbo via in-context learning, and a method for moving from binary to graded bias annotation based on Best-Worst Scaling. As reviewer XADC summarizes, the resulting data is analyzed through 3 lenses: inter-annotator agreement, sample themes and baseline classification tasks, revlealing that annotators overall consider attacks on groups to be more biased than attacks on individuals, and that LLMs such as GPT4 cannot reason well about fine-grained gender bias classification.

The reviewers agree that the paper is overall sound, with adequate description of the methodology and support for the claims.
The paper and data leave open some questions around subjectivity in annotation and the impact of the gender of annotators, which are particularly important from a human-centered NLP perspective. However, the current scope of the work represents an interesting step forward in characterizing gender bias in NLP systems.

I recommend incorporating the discussion of related work, limitations, motivation, and examples that were brought up in the thorough discussion with reviewers. As a minor point, it would be useful to preview the findings of the paper in the abstract, as it currently describes what was done, but leaves the reader wondering what was learned.